# *Brcd1* Is Associated with Plant Height through the Gibberellin Pathway in *Brassica rapa* L.

**Wei Wang [1,†], Junyu Hu [1,†], Bing Fang [2], Xu Gao [1], Chunming Hao [1], Yizhuo Mu [3], Hui Feng [1], Gaoyang Qu [1,*] and Yugang Wang [1,*]**

1 College of Horticulture, Shenyang Agricultural University, Shenyang 110866, China
2 Foreign Language Teaching Department, Shenyang Agricultural University, Shenyang 110866, China
3 College of Agriculture and Environmental Science, University of California Davis, Davis, CA 95618, USA
* Correspondence: qgy@syau.edu.cn (G.Q.); ygwang@syau.edu.cn (Y.W.); Tel.: +86-24-88487143 (Y.W.)
† These author contributed equally to this work.

**Abstract:** In crops such as *Brassica rapa*, the agronomic trait plant height affects the leaf light absorption rate, benefits substance accumulation, and increases production by increasing the biological yield of the population. The mechanism of plant height was adequately studied in *Brassica napus* and *Arabidopsis*, while *Brassica rapa* had been rarely reported. Herein, we studied a *B. rapa* dwarf mutant *Brcd1*[YS]. Compared with its wild-type, Yellow sarson, *Brcd1*[YS] showed shorter hypocotyls and slow growth, with inhibited root elongation and reduced numbers of lateral roots. Chlorophyll content determination and pollen viability detection indicated that the mutant's chlorophyll content was higher than that of the wild-type; however, its pollen was inactive. Cytological identification showed that the number of cells in *Brcd1*[YS] leaves was significantly reduced and were arranged irregularly compared with those of the wild-type. Bulked Segregant RNA sequencing combined with conventional linkage mapped the dwarf mutation *Brcd1*[YS] to Chromosome A06, at position 21,621,766 to 24,683,923 bp. Application of exogenous gibberellic acid (GA) partially rescued the dwarf phenotype of *Brcd1*[YS]. GA-related genes *BraA06g034110.3C* and *BraA06g033010.3C* were identified as the most promising candidate genes. These results make a valuable contribution to our understanding of the mechanism of plant height determination in *B. rapa*, paving the way for further breeding of dwarf *B. rapa*.

**Keywords:** dwarf; *Brassica rapa*; gibberellin; mapping





## 1. Introduction

*Brassica rapa* is an economically important vegetable that has long been cultivated in China and Southeast Asia. Among *B. rapa* strains, Chinese cabbage and Pak-choi are used mainly as vegetable foodstuffs, whereas turnip rape and yellow sarson are commonly used to produce oilseed and fodder [1]. *B. rapa* is one of the oil crops in the world, which show easy lodging and poor stress resistance, resulting in significant production losses. The yield of most *B. rapa* varieties is affected by plant density and lodging, which also restricts mechanical harvesting, particularly for hybrids. Meanwhile, plant height is likewise extremely important for improving the ecological adaptability and disease resistance of *B. rapa*. Dwarfing is an important factor affecting lodging plant structure and plant density; thus, dwarf breeding has always been an important goal for breeders [2,3]. There are few studies on the mechanism of plant height in *B. rapa*, and the regulatory locus found are also very limited.

Plant height is affected by phytohormones, such as gibberellin (GA), brassinosteroid (BR), and auxin (IAA). GA plays a vital role for controlling plant height, and the majority of plant height-related genes that have been utilized in agriculture are GA biosynthesis or response pathway genes [4,5]. GA is also a key factor that regulates the whole growth

process of plants, from seed germination and stem and root extension to flowering and fruit ripening [6]. Most of the dwarfing mutants identified in plants were related to the GA pathway, and the effect of GA on plant height can be divided into two types: mutation of the GA signal transport pathway and mutation of gene-encoding enzymes in the GA synthesis pathway. In the former case, exogenous GA did not change the phenotype, whereas in the latter case, exogenous GA could restore the phenotype of a dwarf mutant [7,8]. The semi-dwarf and high yield varieties developed during the "Green revolution" were associated with a significant change in plant hormone GA biosynthesis and signaling pathways, and the most representative dwarf genes were rice sd1, and wheat *Rht-B1b* and *Rht-D1b* [9,10]. Rice sd1 encodes a defective GA biosynthetic enzyme, GA20ox, which reduces the GA content in the sd1 mutant. Wheat *Rht-B1b* and *Rht-D1b* encode a DELLA protein, which negatively regulates the GA signal pathway, thus leading to a dwarf phenotype [11,12].

Recently, an increasing number of genes associated with dwarf phenotypes were identified and isolated in Brassica plants, which paved the way for deciphering the molecular mechanism of dwarfing. Dwarfing can influence Brassica plant architecture and yield, indeed, knockout of two *BnaMAX1* homologs and the *BnaA03.BP* gene using CRISPR/Cas9-targeted mutagenesis in *Brassica napus* led to a semi-dwarf phenotype and increased production [2,13]. In the GA pathway, *B. napus DS-3* (encoding a DELLA protein involved in the GA pathway) was mapped to chromosome C07, the overexpression of which can result in a dwarf phenotype [14]. In *B. rapa*, *dwarf2* is homologous to the *A. thaliana RGA* gene, both of which encode DELLA proteins that affect GA signal transduction pathway, eventually leading to a dwarf phenotype [15]. GA2ox6 overexpression in *B. napus* significantly reduced plant height and the development of the stem and hypocotyl, whereas chlorophyll accumulation was enhanced [16]. In another pathway, the *B. napus* auxin-related dwarf gene, *ds-4*, negatively regulates plant height through an auxin signaling pathway; *BnaA03.IAA7* and *BnaARF6/8* also regulated stem elongation and plant height through the auxin pathway [17,18]. Two genes, *BraA01000252* and *BraA05004386*, were identified in Chinese cabbage, which, through the upregulation of novel_15 and novel_54, might lead to the inhibition of BR synthesis, resulting in plant dwarfing [19]. In summary, most dwarf studies in Brassica plants were related to the GA and IAA pathways; however, the specific molecular mechanisms require further exploration.

Previously, we used dwarf mutant *Brcd1* and normal plant height inbred line Yellow Sarson (YS) as the donor and recipient parents, respectively. Bulk segregant RNA sequencing (BSR-Seq) combined with conventional mapping, employing the $F_2$ population derived from the cross between *Brcd1* and YS primary mapped *Brcd1* to chromosome A06. Sequencing, expression analysis, and exogenous gibberellin spraying allowed the prediction of two GA-related genes as candidate genes for *Brcd1*. These results will promote further studies of the molecular mechanism governing plant height in *B. rapa*.

## 2. Materials and Methods

### 2.1. Plants and Measurement of Their Traits

RcBr (Rapid Cycling *B. rapa*) is an extremely early flowering, normal plant height inbred line, which was used for EMS mutagenesis to generate the dwarf mutant *Brcd1* (*B. rapa dwarf mutant1*). The $F_2$ *Brcd1/Yellow sarson* (YS) population was subsequently developed using self-pollination, which were donated by Scoot Woody (University of Wisconsin). One individual with normal plant height, but which was heterozygous at the *Brcd1* locus (identified by two markers, L06-09 and L06-14, on chromosome A06) in the $F_2$ population was selected for three successive backcrosses with the recurrent parent YS and one generation of selfing, according to the method detailed in our previous study [20]. Individuals with a dwarf phenotype were selected and named $Brcd1^{YS}$, which should be near-isogenic lines (NILs) for the recurrent parent, *Yellow sarson*.

Individual plant phenotypic scores were calculated to assess plant growth indices, such as the plant height, distance from the plant base to the stem apex in five stages (the first true leaf, third true leaf, bolting stage, five days after bolting, and ten days after bolting),

leaf length, and leaf width, were measured every five days from the third true leaf rosette stage to growth for 45 days. Root evaluations include measurement of their length and number of main and lateral root for 45 days of growth. $Brcd1^{YS}$ and YS were evaluated under normal conditions, and every index was analyzed in ten individuals. SPSS v17.0 (IBM Corp., Armonk, NY, USA) was used to determine the means of the phenotypic data using Student's *t*-test.

$Brcd1^{YS}$ was crossed with YS to generate the $F_2$ population. Five hundred $F_2$ individuals grown under greenhouse conditions from March 2019 were used for BSR Seq analysis. Meanwhile, conventional linkage analysis was carried out using 500 $F_2$ individuals in October 2019. All plants were sown directly into 10 cm pots, without any extra vernalization, under growth conditions as described in our previous study [21]. The experiment was conducted at the Experiment Station of Shenyang Agriculture University, Shenyang, China (41.8° N, 123.4° E).

## 2.2. Stem and Leaf Cell Morphological Characterization

To characterize their morphology, stems and true leaves from mutant $Brcd1^{YS}$ and YS at the same growth period made into paraffin sections for examination. A fresh stem and leaf from mutant $Brcd1^{YS}$ and YS were put it into separate bottles containing penicillin and fixed in formaldehyde, acetic acid and ethanol (FAA), followed by further vacuum storage for 24 h at 25 °C. Different concentrations of alcohol (30–95%) were used to dehydrate the samples for 2 h, with a final incubation in 100% alcohol overnight at 4 °C, followed by xylene permeation. Finally, the samples were paraffin-embedded and sectioned using a microtome (LeicaRM2016, Wetzlar, Leica, Germany). An optical microscope (Nikon ECLIPSE 80i, Tokyo, Japan) was used to observe the shape of the stem and leaf. The NIS-Element SF3.2 software ( Nikon, Tokyo, Japan)was applied to conduct the photo, select the adjusted image, and use the capture option to save.

## 2.3. Photosynthetic Index Determination

To identify the growth difference of $Brcd1^{YS}$ and YS, we measured the photosynthetic rate ($P_n$), the intercellular $CO_2$ concentration ($C_i$), the transpiration rate ($E$), stomatal conductance ($G_s$), the photosynthetic pigment content (chlorophyll a, chlorophyll b, chlorophyll, carotenoid), pollen viability, and stem cell morphology.

After the fourth true leaf of the plant was fully unfolded, mutant $Brcd1^{YS}$ and YS in the same period were carried out on a sunny day between 08:00–12:00 h. The intensity of plant illumination was 800 μmol m$^{-2}$s$^{-1}$, $CO_2$ concentration in cuvette of gas analyzer approximately 377.37–396.89 μmL, $H_2O$ concentration in cuvette of gas analyzer around 19.61–28.05 mmL, and the fourth true leaf was used to identify $P_n$, $C_i$, $E$ and $G_s$ using a Li-6400 portable photosynthesis system (LI-COR Biosciences, Lincoln, NE, USA); three replicates were performed, with six plants in each replicate. The photosynthetic indexes were measured with a modified from Li et al. (2019) [22]. Image J 1.8.0.345 software (National Institutes of Health, Rockville Pike Bethesda, America) was used to analyze the cell number.

## 2.4. Chlorophyll Content Determination

After the third true leaf of the plant was fully unfolded, mutant $Brcd1^{YS}$ and YS in the same period were selected on a sunny morning, the third true leaf was used to evaluate the photosynthetic pigment content. Ten 0.1 cm-diameter round leaves from $Brcd1^{YS}$ and YS with mix solution were placed in the dark for 24 h. After the leaf had completely faded, an ultraviolet spectrophotometer (DU-800; Beckman Coulter, Brea, CA, USA) was used to measure the light absorption value at wavelengths 663 nm, 645 nm, and 440 nm, respectively. The chlorophyll and carotenoid contents were measured with a modified from Arnon et al. (1949) [23]. The formulae are summarized below:

$$Chla = (12.72OD663 - 2.59OD645) \, V/1000 \, W;$$

$$\text{Chlb} = (22.88\text{OD645} - 4.67\text{OD663})\,\text{V}/1000\,\text{W};$$

$$\text{Chl(a + b)} = (8.05\text{OD663} - 20.29\text{OD645})\,\text{V}/1000\,\text{W};$$

$$\text{Car} = (1000\text{OD440} - 3.27\text{Chla} - 104\text{Chlb})\,\text{V}/229 \times 1000\,\text{W}.$$

Note: OD represents the optical density corresponding to the measured wavelength; V represents the total volume of chlorophyll extract (mL); W represents material quality (g).

### 2.5. Pollen Viability Determination

Acetic red staining was used to evaluate the pollen viability of mutant *Brcd1*^YS and YS, according to the method by Tan et al. [24]. The anthers were extracted in the flowering stage from mutant *Brcd1*^YS and YS; pollen was placed on a slide with 1% magenta acetate and observed by Inversed Fluorescent Microscope (OLYMPUS DP80, Olympus Corporation, Tokyo, Japan). Analysis of each sample was repeated three times.

### 2.6. RNA Isolation and Extreme Pool Construction

A D-pool (dwarf pool) with 40 dwarf individuals and an N-pool (normal pool) with 40 normal plant height individuals were constructed as two extreme pools for BSR-seq. At 45 days after sowing, according to the phenotypes of 500 $F_2$ individuals, the two sets of 40 extreme phenotype plants were selected, and their true leaves were sampled to isolate RNA. The TRIzol reagent (Invitrogen, Carlsbad, CA, USA) was adopted to extract the total RNA. An Agilent 2100 Bioanalyzer (Agilent Technologies, Palo Alto, CA, USA), a NanoDrop spectrophotometer (Thermo Fisher Scientific Inc., Waltham, MA, USA), and 1% agarose gel were used to assess the quantity and quality of the extracted RNA. RNA (1 μg) with an RNA integrity number (RIN) > 7 was then processed for next-generation sequencing library construction (NEBNext® Ultra™ RNA Library Prep Kit for Illumina®; NEB, Ipswich, MA, USA). The methods of RNA extraction and quality determination were the same as those detailed by Qu et al. [25].

### 2.7. BSR-seq

After the cDNA libraries were constructed, the Illumina HiSeq 2500 platform (Illumina, San Diego, CA, USA) was used to sequence them. Data analysis was performed by GENEWIZ company (Suzhou, China). To ensure the accuracy of the data, low-quality bases were removed using Trimmomatic v0.30, (Illumina, CA, USA) to generate clean data [26]. The clean data were compared with the *B. rapa* reference genome V3.0 (BRAD; http://brassicadb.cn/#/Download/ accessed on 10 May 2020) using Hisat2 (Johns Hopkins University, Baltimore, MD, USA) [27]. Single Nucleotide Variation (SNV) was detected using Samtools v0.1.18 (Trust Sanger Institute, Cambridgeshire, UK) and Bcftools v0.1.19 (Trust Sanger Institute, Cambridgeshire, UK) [28,29]. The Euclidean distance (ED) value was calculated based upon an mpileup file, which was generated by samtools v0.1.18 (Trust Sanger Institute, Cambridgeshire, UK) for BSR-seq. According to the basic principle of the ED value, the occurrence frequency of the four bases A, T, C, and G at the SNV site was statistically different in the population, and the corresponding base frequency of the two trait groups was calculated by distance. To eliminate the background noise, the ED value of each different SNV site was raised to the power of five, termed ED^5 [30]. Screening identified the top 1% of ED^5 values as a threshold value, which was compared with different SNV sites distributed in different chromosomes, finally confirming the candidate region for the dwarf phenotype.

### 2.8. Extraction of DNA and the Development of Markers

The cetyltrimethylammonium bromide (CTAB) method [31], with minor modifications, was used to extract total DNA. PCR was then carried out as described previously [21]. The specific approaches were summarized as follows: the dry and fresh plant leaves were well ground, 700 μL of CTAB solution was added, followed by 700 μL of chloroform isoamyl alcohol and frozen absolute ethanol, frozen at −80 °C for 8 min, eluted with 70% ethanol



solution and dried overnight, and finally 100 μL of sterilized water was added. Specific primers and insertion/deletion (InDel) markers were designed based on the sequence variations from BSR-seq data between *Brcd1*[YS] and YS using Primer Premier 5.0 software (Premier Biosoft, San Francisco, CA, USA). In addition, some simple sequence repeat (SSR) markers were employed, which had been used previously in our laboratory. All the polymorphic markers are listed in Table S2.

### 2.9. Preliminary Localization of Brcd1 Dwarfing Mutation

Linkage analysis was used to confirm the candidate dwarfing region identified using BSR-seq, with the aid of polymorphic markers. Linkage analysis used an $F_2$ population comprising 80 homozygous recessive individuals that were sown in September 2019. Six and eight polymorphic markers were developed in the A06 and A08 chromosomes, respectively. Progeny of recombinant individuals (1500 individuals) were sown in the greenhouse in October 2019 and used to narrow down the candidate region.

### 2.10. Exogenous GA₃ Treatment and Candidate Gene Annotation Analysis

A total of 15 mutant *Brcd1*[YS] and 15 YS plants were treated using exogenous $GA_3$, with distilled water as the control treatment. The $GA_3$ powder (Solarbio, Beijing, China) was dissolved in 5 mL of ethanol and diluted with distilled water to 500 mg/L. The $GA_3$ solution and distilled water were then sprayed on the individuals with fully expanded cotyledons. The spray was applied six times every other day, each treatment had three replications, and each replication comprised five individuals. The method was carried out as previously described by Gao et al. [32].

Gene functional analyses were carried out at The Arabidopsis Information Resource (TAIR; http://www.arabidopsis.org/, accessed on 10 May 2020 ) and The Brassica rapa database (BRAD; http://brassicadb.cn/#/Download/, accessed on 10 May 2020). The combined annotation information of both resources were used to predict the candidate genes in the identified region.

### 2.11. Candidate Gene Expression Determination

Quantitative real-time reverse transcription PCR (qRT-PCR) was used to detect the expression levels of candidate genes. Total RNA of *Brcd1*[YS] and $GA_3$-treated *Brcd1* (*Brcd1*[YS-GA]$_3$) from roots, leaves, and stems were extracted using an RNA extraction kit (Aidlab Biotechnologies Co., Ltd., Beijing, China), followed by reverse transcription into cDNA (TIANGEN, Beijing, China). The cDNA formed the template for the qPCR step of the qRT-PCR protocol, which was carried out using SYBR Green PCR Master Mix (Takara, Dalian, China), following the manufacturer's instructions. The relative expression levels were determined using the $2^{-\Delta\Delta Ct}$ method and each sample had three independent technical replicates [33]. The actb gene (encoding beta actin) formed the internal control. Specific primers were designed using Primer Premier 5.0 (Premier Biosoft, Palo Alto, CA, USA) and are shown in Table S2. QuantStudio™6 Flex Manager software (ABI, Los Angeles, CA, USA) was used to analyze the data [34,35].

### 2.12. Sequence Analysis of Candidate Genes

Candidate genes obtained from the *Brassica rapa* genome were used to design specific PCR primers (Table S2) with Primer Premier 5.0 to amplify the full-length sequences. The PCR products were purified using a Gel Extraction Kit (CWBIO, Beijing, China), cloned into the pGEM T Easy Vector (Promega, Madison, WI, USA), and then sequencing was performed at Sangon Biotech (Shanghai, China). Sequence alignment was carried out using DNAMAN software (Lynnon Biosoft, San Ramon, CA, USA).

## 3. Results

### 3.1. Phenotypic Characterization of the Dwarf Mutant

To explore the growth variation of dwarfing, we identified the phenotypic characteristics of the mutant *Brcd1*[YS] by comparing it with YS. *Brcd1*[YS] showed significant dwarfing compared with YS, grew slowly, and had very short hypocotyls. In addition, *Brcd1*[YS] leaves showed a dark color, prickly hairs, a narrow leaf area, and prominent veins (Figure 1). *Brcd1*[YS] showed a significant difference in plant height, leaf length, and leaf width compared with those of YS (Figure 2 and Table S1). *Brcd1*[YS] root elongation was significant restrained, and the number of lateral roots and main roots also decreased significantly compared with those in YS (Figure 3).

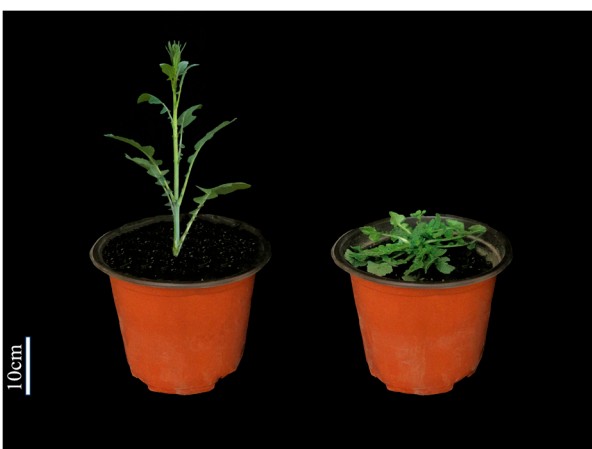

**Figure 1.** Phenotypic characterization of *Brcd1*[YS] and YS. The dwarf mutant '*Brcd1*[YS]' (**right**) and wild-type 'YS' (Yellow Sarson, **left**) grown under natural conditions. Scale bars = 10 cm.

### 3.2. Analysis of Physiological Indices

Our study evaluated the $P_n$, $C_i$, $E$, $G_s$, photosynthetic pigment content (chlorophyll a, chlorophyll b, chlorophyll, carotenoid), pollen viability, and stem cell morphology between YS and *Brcd1*[YS]. The results (Table 1) showed that for *Brcd1*[YS], $P_n$ and $G_s$ were higher than in YS, whereas $C_i$ and $E$ were lower, which indicated that the stress resistance of *Brcd1*[YS] is better than that of YS. The chlorophyll content in *Brcd1*[YS] was higher than that of YS. The pollen levels of mutant *Brcd1*[YS] was significantly decreased and the pollen grains were irregular in shape (Figure 4).

### 3.3. Identification Stem and Leaf Cell Morphologies of the Brcd1[YS] Mutant

The shorter stem resulted in the dwarf phenotype of mutant *Brcd1*[YS], which might have been caused by a decrease in cell number or a shorter cell longitudinal length. To explain the mechanism of shortened stem of *Brcd1*[YS], paraffin section analysis was performed on the stem of *Brcd1*[YS] and YS. Compared with YS, the space in the middle column cells was larger in the stem. Longitudinal section analysis showed that *Brcd1*[YS] parenchymal cells were significantly larger (Figure 5A–D). In the magnified stem section, the number of cells in *Brcd1*[YS] was 27 under the total cell area of 130,081 μm², while that in YS was 55 under the total cell area of 1,266,445 μm² (Table S3). In leaves, the cells of *Brcd1*[YS] had an irregular arrangement and enlarged cell size (Figure 5E–H). In the magnified leaf section, the cell number of *Brcd1*[YS] was 229 when the total cell area was 142,714 μm², while that of YS was 313 when the total cell area was 158,677 μm² (Table S3). Thus, the decrease in cell number and their irregular arrangement might be the major reason for the dwarf phenotype and altered leaves.

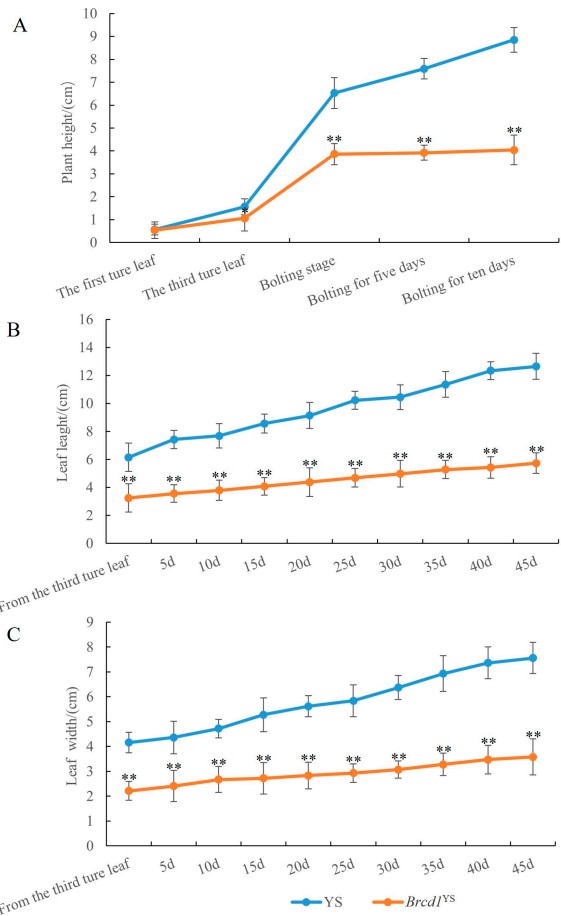

**Figure 2.** Phenotype characterization of *Brcd1*<sup>YS</sup> and YS. The plant height, leaf length, and leaf width were used for the phenotypic characterization of *Brcd1*<sup>YS</sup> and YS, all the traits presented a normal distribution. (**A**): plant height, (**B**): leaf length, (**C**): leaf width. Data are shown as presented the mean ± SD (*n* = 10 plants). ** *p* < 0.01.

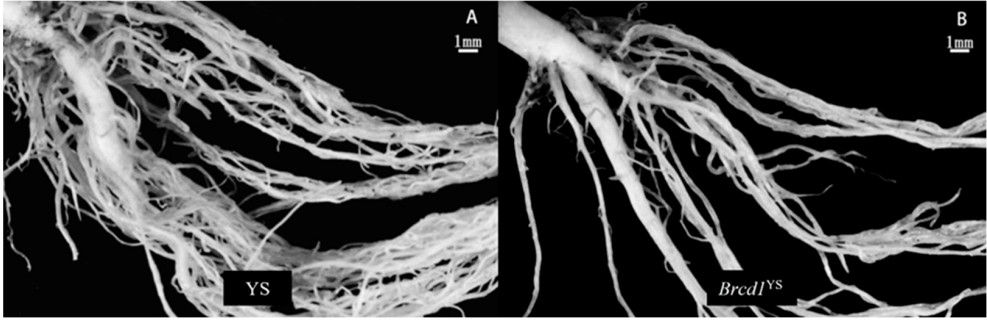

**Figure 3.** Evaluation of the roots of *Brcd1*<sup>YS</sup> and YS. Images of the roots of *Brcd1*<sup>YS</sup> and YS. (**A**): YS, (**B**): *Brcd1*<sup>YS</sup>. Scale bars = 1 mm.

**Table 1.** Evaluation of the physiological indices of YS and *Brcd1*<sup>YS</sup>.

| | Photosynthetic Rate ($\mu$mol m$^{-2}$s$^{-1}$) | Intercellular $CO_2$ Concentration ($\mu$mol mol$^{-1}$) | Transpiration Rate (mmol m$^{-2}$s$^{-1}$) | Stomatal Conductance (mmol m$^{-2}$s$^{-1}$) | Chlorophyll a (mg g$^{-1}$ FW) | Chlorophyll b (mg g$^{-1}$ FW) | Total Chlorophyll (mg g$^{-1}$ FW) | Carotenoid (mg g$^{-1}$ FW) |
|---|---|---|---|---|---|---|---|---|
| YS | 20.44 ± 1.49 | 404 ± 4.05 | 6.17 ± 0.30 | 6.17 ± 16.15 | 1.61 ± 0.01 | 0.47 ± 0.03 | 2.08 ± 0.04 | 0.52 ± 0.03 |
| *Brcd1*<sup>YS</sup> | 25.91 ± 0.74 * | 390 ± 3.31 * | 5.04 ± 0.28 * | 7.69 ± 25.0 * | 1.67 ± 0.01 * | 0.55 ± 0.03 * | 2.20 ± 0.02 * | 0.46 ± 0.02 * |

Note: * *p* < 0.05.

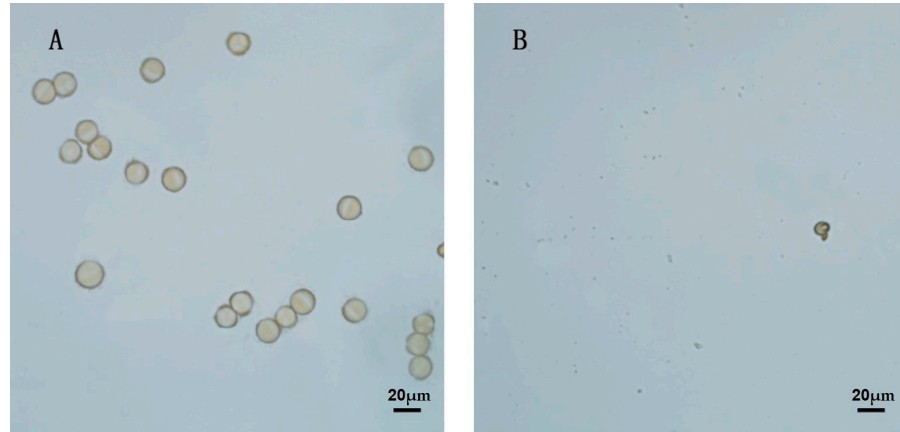

**Figure 4.** Pollen viability determination of *Brcd1*<sup>YS</sup> and YS. Pollen viability determination of *Brcd1*<sup>YS</sup> and YS. (**A**): YS, (**B**): *Brcd1*<sup>YS</sup>. Scale bars = 20 μm.

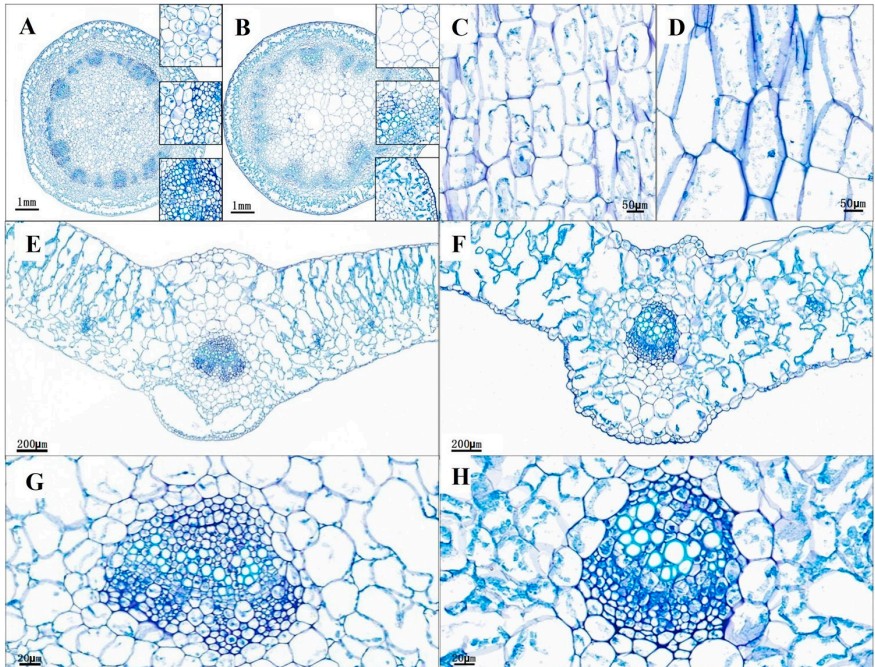

**Figure 5.** Stem and leaf cell morphology of *Brcd1*<sup>YS</sup> and YS. Paraffin sections were used to identify the stem and leaf cell morphologies of *Brcd1*<sup>YS</sup> and YS. (**A**): Stem cross section of YS; (**B**): stem cross section of *Brcd1*<sup>YS</sup>, scale bars = 1 mm. (**C**): Magnified stem section of YS; (**D**): magnified stem section of *Brcd1*<sup>YS</sup>, scale bars = 50 μm. (**E**): Leaf section of YS; (**F**): leaf section of *Brcd1*<sup>YS</sup>, scale bars = 200 μm. (**G**): Magnified leaf section of YS; (**H**): magnified leaf section of *Brcd1*<sup>YS</sup>, scale bars = 20 μm.

*3.4. Primary Mapping and Validation of Brcd1 Locus*

BSR-seq analysis allowed us to map 52,654,592 (D-pool) and 48,289,914 (N-pool) clean reads to the *B. rapa* reference genome. Compared with the *B. rapa* reference genome, we identified 227,846 single-nucleotide polymorphisms (SNPs) in the D-pool and 216,068 SNPs in the N-pool Using the top 1% of the $ED^5$ values as the threshold and the distribution of SNV sites, the candidate regions for the *Brcd1*<sup>YS</sup> were located at chromosome A04 (starting at 17,334,001 and ending at 18,939,299) and A06 (starting at 19,356,018 and ending at 25,157,303); thus, the candidate regions comprised about 6.86 Mb (Figure S1). The candidate gene responsible for the dwarf phenotype was termed *Brcd1*<sup>YS</sup>.

BSR-seq analysis identified two candidate regions for *Brcd1*$^{YS}$; therefore, to validate which one is reliable, 100 homozygous recessive individuals from the F$_2$ population were used to determine the linkage related to the dwarf phenotype. We identified 14 polymorphic markers (Table S2) in the candidate intervals (Chromosomes A04 and A06), which were used for linkage analysis. Eight markers were identified as linked to the *Brcd1*$^{YS}$ in Chromosomes A06, and six markers were identified as not linked in Chromosome A04; thus, *Brcd1*$^{YS}$ was primarily mapped to Chromosome A06. Furthermore, using six polymorphic markers (Table S2) from the candidate region from chromosome A06, 500 homozygous-recessive individuals of F$_2$ populations were typed to fine map *Brcd1*$^{YS}$. Positioning of recombinant individuals allowed *Brcd1*$^{YS}$ to be narrowed down to an interval of 3.06 Mb between markers L06-09 and L06-14 (physical position: 21,621,766 to 24,683,923 bp) on chromosome A06.

### 3.5. Brcd1 Mutant Plants Could Be Rescued by Exogenous GA$_3$ Treatment

Plant dwarfing is usually mediated by the GA pathway; thus, exogenous GA$_3$ was used to spray dwarf plants, with water spraying as the control. The result showed that the dwarfing mutant *Brcd1*$^{YS}$ treated with exogenous GA$_3$ recovered its height; however, there were no significant changes in leaf length, leaf width, and leaf fluctuation degree (Figure 6). Therefore, dwarfing *Brcd1*$^{YS}$ was sensitive to exogenous gibberellin treatment, and it was speculated that the reason for the plant height reduction of *Brcd1*$^{YS}$ was a defect in hormone synthesis.

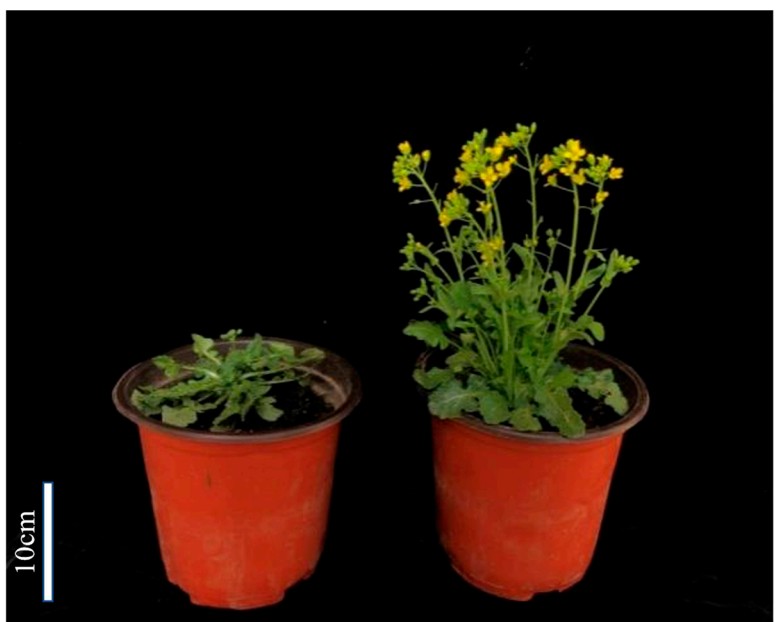

**Figure 6.** Phenotypes of the *Brcd1*$^{YS}$ in response to exogenous gibberellin (GA$_3$) treatment. The dwarf mutant '*Brcd1*$^{YS}$' (**left**) and dwarf mutant '*Brcd1*$^{YS}$' treated with GA$_3$ (**right**) grown under natural conditions. The plants are shown at 45 days after sowing (DAS). Scale bars = 10 cm.

### 3.6. Prediction of Candidate Genes

A total of 548 genes were detected in the 3.06 Mb candidate region (*BraA06g032610.3C*–*BraA06g038080.3C*). Combining the results of exogenous GA$_3$ treatment with the *B. rapa* reference genome and Arabidopsis genome information, only two GA-related genes were identified: *BraA06g033010.3C* and *BraA06g034110.3C*, and the specific information is shown in Table S4. *BraA06g033010.3C* is homologous with *Arabidopsis AT5G25900*, which is involved the GA biosynthetic pathway, and encodes a member of the CYP701A cytochrome p450 family. *BraA06g034110.3C* is homologous with *Arabidopsis AT5G27320* and rice *OsGID1*, which encode a GA receptor with high affinity for GA$_4$, which can interact with DELLA

proteins in the presence of GA$_4$ [36,37]. Thus, the two GA-related genes, *BraA06g033010.3C* and *BraA06g034110.3C*, were identified as promising candidate genes for *Brcd1*$^{YS}$.

### 3.7. Expression Analysis of GA-Related Genes

QRT-PCR was used to detect the expression levels of the candidate genes in the dwarf mutant and sprayed individuals. Two specific primer sets, RT-22 and RT-41, were used to analyze the *BraA06g033010.3C* and *BraA06g034110.3C* genes. The results indicated that that the expression levels of the two genes were significantly upregulated in GA$_3$-treated plants. The expression levels of *BraA06g033010.3C* and *BraA06g034110.3C* were significantly different in the leaf, stem, and root (Figure 7). Thus, *BraA06g033010.3C* and *BraA06g034110.3C* are candidate genes for *Brcd1*$^{YS}$.

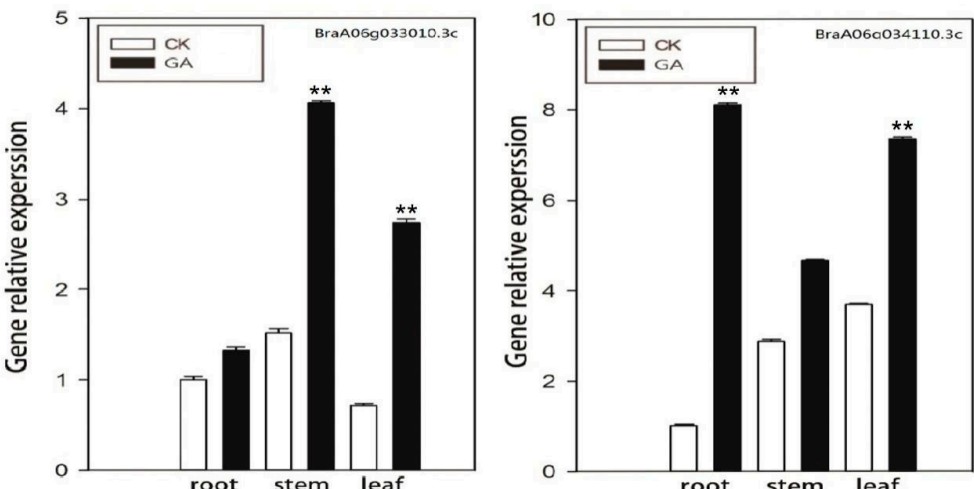

**Figure 7.** Expression patterns of gibberellin (GA)-related genes in plant tissue. Expression levels of *BraA06g033010.3C* and *BraA06g034110.3C* in the leaf, stem and root according to quantitative real-time reverse transcription PCR (qRT-PCR) analysis. Error bars indicate the standard errors (*n* = 3); white is mutant *Brcd1*$^{YS}$ is shown in white and GA treatment is shown in black, ** *p* < 0.01.

### 3.8. Sequence Variation Analysis of Candidate Genes

The sequences of the two candidate genes *BraA06g033010.3C* and *BraA06g034110.3C* from the two parents were obtained after map-based cloning, which were analyzed using DNAMAN V6.0 software (Lynnon BioSoft, Montreal, QC, Canada). The specific primer sets 33010-C and 34110-C were used to detect variations in the *BraA06g033010.3C* and *BraA06g034110.3C* sequences, respectively (Table S2). The sequence of *BraA06g033010.3C* comprises 2195 bp (22,597,175 to 22,599,369 bp), with six exons and five introns. Compared with the coding sequence (CDS) of YS, the CDS of *Brcd1*$^{YS}$ has a C base inserted at base 496, resulting in glutamine becoming lysine (Figure S2). The gene for *BraA06g034110.3C* comprises 1690 bp (23,226,835 to 23,228,525), with four exons and three introns. Compared with that of YS, the CDS of *Brcd1*$^{YS}$ has an A to G change at base 322, which results in threonine becoming alanine (Figure S3). These results further support *BraA06g033010.3C* and *BraA06g034110.3C* as candidate genes for dwarf mutant *Brcd1*$^{YS}$.

## 4. Discussion

In the present study, phenotypic and cell morphology identification of *Brcd1*$^{YS}$ and YS revealed that the *Brcd1*$^{YS}$ exhibited geotropic growth during the entire growth period, and leaf and stem cells showed significant differences. Conventional mapping requires the development of many polymorphic primers and is time consuming. BSR-seq together with classical linkage analysis is a powerful tool to delineate candidate regions that control agronomic traits in a various crop [4,32,38]. In the present study, BSR-seq based on the F$_2$

population derived from mutant *Brcd1* and YS identified the candidate intervals responsible for the dwarf phenotype, which were further confirmed using linkage analysis.

The GA pathway is a major effector for plant height, in which $GA_1$, $GA_3$, $GA_4$, and $GA_7$ are the major bioactive GAs [39]. In our study, exogenous $GA_3$ spraying rescued the phenotype of dwarf mutant, and *BraA06g033010.3C* and *BraA06g034110.3C* expression levels were significantly upregulated in the root, leaf, and stem by $GA_3$, indicating that inhibition of GA synthesis affected plant morphogenesis. *BraA06g033010.3C* and *BraA06g034110.3C* are both key genes in the GA synthesis pathway. *BraA06g033010.3C* is homologous with *Arabidopsis* gene *AT5G25900* and maize gene *ZmGRF* [34], which encode a member of the p450 family that is involved in GA biosynthesis. Studies suggested that *ZmGRF* overexpression in *Arabidopsis* promoted flowering and the flowering time, which was not affected by the photoperiod, and at the same time increased the cell size in leaves and stems, thereby changing the leaf morphology, similar to our study (Figures 2 and 5) [36]. Lyu et al. [40] also identified this phenomenon in *Arabidopsis*. *BraA06g034110.3C* is homologous with *Arabidopsis* gene *AT5G27320* and rice *OsGID1*, encoding a GA receptor gene with higher affinity to $GA_4$, which can also interact with DELLA proteins in the presence of GA4 [28,40]. Griffiths et al. and Iuchi et al. found that loss of function of GID1 resulted in a dwarf phenotype in *Arabidopsis* that was not sensitive to GA, but showed signs of infertility [41,42]. These results contrasted with those of our study, in which exogenous $GA_3$ spraying could restore the phenotype of the dwarf mutant. Our results demonstrated that the dwarf mutant was related to GA synthesis, but whether GA related genes in the candidate region control this phenotype requires validation using transgenic plants.

In this study, *Brcd1*[YS] showed extreme dwarfing, and is one of the few dwarf mutants found in *B. rapa* at present. Our results indicated that *Brcd1* has a significant regulatory effect on the plant height, suggesting that it will be important for dwarf breeding in the future. *B. rapa* is frequently used for feed and oil seeds, and lodging resistance is very important in production [43]. Many studies have shown that dwarfing can increase the yield and improve plant stress tolerance, for example, in rapeseed and rice [44,45]. In addition, it was found that simultaneous knockout of all four *BnaMAX1* homologous genes resulted in semi dwarfing and significantly increased the yield per plant [13]. During the Green Revolution, excessive use of nitrogen fertilizer led to crop collapse, and the development of dwarfing genes could help improve crop yields [9]. In this study, $P_n$, $C_i$, $E$, and $G_s$ analyses showed that the stress tolerance of the plants was significantly improved; however, pollen viability detection indicated that the dwarf mutant was male sterile. Meanwhile, cell morphological observations were performed on the stems and leaves of two parents, *Brcd1*[YS] and YS. It was found that the median column cells and parenchymal cells in the stem were significantly larger than YS in *Brcd1*[YS], and the leaf cells of *Brcd1*[YS] were irregularly arranged. Therefore, we think this may be one of the causes of plant dwarf. Kim et al. (2011) found that the dwarf phenotype exhibited by tyrosine decarboxylase transgenic rice is related to the irregular arrangement of leaf cells [46]; Jiang et al. (2012)'s results showed that the leaf epidermal cells and internode parenchyma cells of dwarf mutation in maize are irregular in shape and are arranged in a more random fashion [47]. Thus, to understand the mechanism of dwarfing, gene knockout using CRISPR-Cas9 could be used to create *B. rapa* with high quality and high yield.

## 5. Conclusions

In conclusion, we mapped a dwarf gene for *B. rapa*, *Brcd1* to a physical interval of 3.06 Mb on chromosome A06. *Brcd1* regulates the dwarf phenotype positively at all growth stages of *B. rapa*. Expression and annotation analysis identified the most likely candidate genes as *BraA06g033010.3C* and *BraA06g034110.3C*, both of which affect the synthesis of GA. Exogenous $GA_3$ spraying could restore the phenotype, which further indicated that *BraA06g033010.3C* and *BraA06g034110.3C* were the most promising genes for the dwarf mutant phenotype. Our results provide a deeper understanding of the mechanisms of

dwarf plant formation and supply a genetic resource for the study of crop improvement in *B. rapa*.

**Supplementary Materials:** The following supporting information can be downloaded at: https://www.mdpi.com/article/10.3390/horticulturae9020282/s1, Figure S1: ED$^5$ values of differential SNPs distributed on *Brassica rapa* chromosomes based on the BSR-Seq data. SNP distributions on *B. rapa* chromosomes. The 10 chromosomes are shown on the x-axis and the ED$^5$ values of the filtered SNP are on the y-axis; the horizontal bar indicates the top 1% threshold value. ED, Euclidean distance; SNP, single nucleotide polymorphism; Figure S2: The gene sequence of *BraA06g033010.3C*; Figure S3: The gene sequence of *BraA06g034110.3C*; Table S1: Identified phenotype mean value of YS and *Brcd1*$^{YS}$; Table S2: Primers sequence; Table S3: Stem and leaf cell number as well as area statistics of YS and *Brcd1*$^{YS}$; Table S4: Candidate gene annotation.

**Author Contributions:** Y.W. designed the experiments. J.H. and W.W. conducted the experiments, G.Q. wrote the manuscript, and performed the data analysis, X.G., Y.M., H.F., B.F. and C.H. assisted with the data analysis. G.Q., B.F. and Y.W. revised the manuscript. All authors reviewed and approved this submission. All authors have read and agreed to the published version of the manuscript.

**Funding:** This research was supported by the National Natural Science Foundation of China (grant number 32072569).

**Data Availability Statement:** The data used in this study are available from the author on reasonable request.

**Acknowledgments:** We would like to thank Scoot Woody (University of Wisconsin) for kindly providing the dwarf mutant seed.

**Conflicts of Interest:** The authors declare that they have no conflict of interests.

**Ethical Statement:** The authors note that this research was performed and reported in accordance with ethical standards of scientific conduct.

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
