# Peer review of "Brcd1 Is Associated with Plant Height through the Gibberellin Pathway in Brassica rapa L."

_horticulturae, doi:10.3390/horticulturae9020282_

Round 1

Reviewer 1 Report

The topic of this manuscript is valuable as Brassica rapa and its growth is important for agriculture and some industrial aspects.

The manuscript is well designed however an English revision is needed, because of many mistakes in English grammar.

Introduction provides many details of the background of this study but I think the last part of it is the same as we can see in the Mat and Meths, so please eliminate it or rewrite it.

In the Mat and Meths, the Section 2.2 is a little short as it describes many methods, so please separate them and write alone the photosynthetic parameters and the chlorophyll measurements with separate protocols and citations.

I think Section 2.3 with morphology should be earlier than biochemical or photosynthetical measurements.

In the L 136, please add how the photo documentation was conducted.

In the Section 2.4 what authors mean in the L 142 “under a microscope”? Please add the precise type of that.

In the Results, at Fig 2. I cannot see the standard deviations in the diagrams, please add to the columns.

My question is that authors measured the length of PR roots or LR numbers? It is only a photo.

At Table 1, please adjust the name of the parameters and what is the chlorophyll mean? The total chlorophyll?

At Fig 4., is there any data or percentage about the number of pollens? Please add it.

In the case of 3.3 identification morphologies, please separately discuss these results.

In the Fig 5., is there any measurements of these parameters eg. by ImageJ analysis, It would be interesting to compare the anatomy of these leaves and roots.

At Fig.6, the numbers and letters are very small, please add bigger of them.

I think Fig.6, should be placed to the Supplements.

At 3.8, sequence variation analysis, please add the software version number of DNAMAN.

Discussion is well written, but please eliminate “E” from CRISPER-Cas9.

Author Response

Dear reviewer,

Thank you for your letter concerning our manuscript entitled "Brcd1 is associated with plant height through the Gibberellin pathway in Brassica rapa L" (Manuscript Number: horticulturae-2196183). The reviewers’ comments are valuable and helpful for revising and improving our paper. We have studied the comments carefully and made corrections accordingly which we hope meet with approval. Point-by-point responses are provided below and changes within the revised manuscript are highlighted in yellow.

Upon your review of our revised manuscript, we hope you will find it acceptable for publication in Horticulturae, and look forward to your response.

Yours sincerely,

Gaoyang Qu and Yugang Wang

Response to Reviewer #1:

We have revised the manuscript in native English speaking scientists of Elixigen Company (Huntington Beach, California).  

Comment 1: Introduction provides many details of the background of this study but I think the last part of it is the same as we can see in the Mat and Meths, so please eliminate it or rewrite it.

Response:

Thanks for your valuable suggestions and we have rewrote the last part of the introduction. See page 2, line 80-86.

Comment 2: In the Mat and Meths, the Section 2.2 is a little short as it describes many methods, so please separate them and write alone the photosynthetic parameters and the chlorophyll measurements with separate protocols and citations.

Response: Thanks for your valuable suggestions. We separated the section 2.2 and write alone (section 2.3 and 2.4 in revised manuscript), meanwhile the photosynthetic parameters and the chlorophyll measurements with separate protocols and citations. See page 3-4, section 2.3 and 2.4, line 127-155.  

Comment 3: I think Section 2.3 with morphology should be earlier than biochemical or photosynthetical measurements.

Response: Thanks for your valuable suggestions. In order to be more logical, we reordered the section 2.3 and 2.2. See Page 3, section 2.2, line 115.

Comment 4: In the L 136, please add how the photo documentation was conducted.

Response: In this study, optical microscope was used to observe the shape of the stem and leaf, and NIS-Element SF3.2 software was used to conduct the photo. The method as follows: select the adjusted image and use the capture option to save. See Page 3, line 124-126.

Comment 5: In the Section 2.4 what authors mean in the L 142 “under a microscope”? Please add the precise type of that.

Response: Sorry for our ambiguous expression. In the L 142 “under a microscope” means “observed by microscope”, the precise type of microscope was Inversed Fluorescent Microscope (OLYMPUS DP80,Japan). We have modified accordingly in our revised manuscript. See Page 4, line 160.

Comment 6: Line 378: In the Results, at Fig 2. I cannot see the standard deviations in the diagrams, please add to the columns.

Response: We have modified accordingly in our revised manuscript and redrew the Fig 2. See page 7, Fig. 2.

Comment 7: My question is that authors measured the length of PR roots or LR numbers? It is only a photo.

Response: In our study, we measured the length of the primary root and the number of lateral roots, the specific differences are clearly observed in the picture (Figure 3), but we not measured specific numbers. Sorry for our carelessness.

Comment 8: At Table 1, please adjust the name of the parameters and what is the chlorophyll mean? The total chlorophyll?

Response: At Table 1, Chlorophyll means total chlorophyll, we have modified accordingly in our revised manuscript. See page 8, Table 1.

Comment 9: At Fig 4., is there any data or percentage about the number of pollens? Please add it.

Response: At Fig 4., pollen viability determination of Brcd1YS and YS found the normal plant YS lots of plump pollen grains, but only a shriveled pollen grain in Brcd1YS. There have no data or percentage about the number of pollens, because no pollen was found in Brcd1YS from the other replicates.

Comment 10: In the case of 3.3 identification morphologies, please separately discuss these results.

Response: According to the result of section 3.3, we separate discussed these results in result part, the according modification in our revised manuscript. See page 12, line 411-419.

Comment 11: In the Fig 5., is there any measurements of these parameters e.g. by Image J analysis, It would be interesting to compare the anatomy of these leaves and roots.

Response: Thanks for your valuable suggestions. The image J software was used to analyze data from Figure 5, the according modification in our revised manuscript. See page 8, line 283-288. Specific information was list in Table S3.

Comment 12: At Fig.6, the numbers and letters are very small, please add bigger of them.I think Fig.6, should be placed to the Supplements.

Response: Thanks for your valuable suggestions. We adjusted the quality of the Fig.6 and placed to the supplementary material Figure S1.

Comment 13: At 3.8, sequence variation analysis, please add the software version number of DNAMAN.

Response: At 3.8, the DNAMAN V6.0 software was used to analysis the sequence variation. See page 11, line 358.

Minor corrections

Discussion is well written, but please eliminate “E” from CRISPER-Cas9.

Response: Sorry for our carelessness, we have modified accordingly in our revised manuscript. See page 12, line 419-420.

Reviewer 2 Report

The manuscript by Wang et al. considers some details in gibberellin pathway of Brassica rapa. This work can be interesting for fundamental investigation of plant hormones. However, I have some remarks:

(1)    The aim, problem, and novelty of investigation should be described in more detail.

(2)    The methods should be described in more detail. Particularly,

-          The scheme of measurement of photosynthetic rate should be explained including time of plant dark adaptation, intensity and time of plant illumination, CO2 and H2O concentration in cuvette of gas analyzer.

-          The description of procedure of pigments estimation should be extended.

-          The description of morphological measurements of plants should be extended.

-          What was the pollen viability criterion?

-          The description of RNA isolation and extreme pool construction should be extended, all instruments should be added.

-          The description of methods of calculation in section “2.6.BSR-seq” should be extended.

-          The description of methods in section “Extraction of DNA and the development of markers” should be extended.

(3)    What are bars in Figure 1 and Figure 7?

(4)    Standard deviations (or errors) and significances should be added in Figure 2.

(5)    Why only one pollen unit was showed in Figure 4B?

(6)    Parameters of cells in Figure 5 should be quantitively described.

(7)    The quality of Figure 6 should be improved.

Author Response

Dear reviewer,

Thank you for your letter concerning our manuscript entitled "Brcd1 is associated with plant height through the Gibberellin pathway in Brassica rapa L" (Manuscript Number: horticulturae-2196183). The reviewers’ comments are valuable and helpful for revising and improving our paper. We have studied the comments carefully and made corrections accordingly which we hope meet with approval. Point-by-point responses are provided below and changes within the revised manuscript are highlighted in yellow.

Upon your review of our revised manuscript, we hope you will find it acceptable for publication in Horticulturae, and look forward to your response.

Yours sincerely,

Gaoyang Qu and Yugang Wang

Response to Reviewer #2

Comment 1: The aim, problem, and novelty of investigation should be described in more detail.

Response: Thanks for your valuable suggestion. We added the aim, problem, and novelty of investigation in abstract and introduction. See page 1-2, line 15-16, 38-43. 

Comment 2: The methods should be described in more detail. Particularly,

-The scheme of measurement of photosynthetic rate should be explained including time of plant dark adaptation, intensity and time of plant illumination, CO2 and H2O concentration in cuvette of gas analyzer.

-The description of procedure of pigments estimation should be extended.

-The description of morphological measurements of plants should be extended.

-What was the pollen viability criterion?

-The description of RNA isolation and extreme pool construction should be extended, all instruments should be added.

-The description of methods of calculation in section “2.6.BSR-seq” should be extended.

-The description of methods in section “Extraction of DNA and the development of markers” should be extended.

Response: Thanks for your valuable suggestions. The methods (photosynthetic rate, procedure of pigments estimation, morphological measurements, RNA isolation, BSR-seq, Extraction of DNA and the development of markers) mentioned above have been improved and supplemented, we have modified accordingly in our revised manuscript. See page 3-4, line 133-140, 148-155, 124-126, 167-173, 182-188, and 194-197.

 In this study, acetic red staining was used to identify the pollen viability. The pollen viability criterion was stained pollen grains are alive, but empty or deformed pollen grains regard as dead and sterile.

Comment 3: What are bars in Figure 1 and Figure 7?

Response: Sorry for our carelessness. We add the bars in Figure 1 and Figure 7, the according modification in our revised manuscript. See page 6, Figure 1. See page 10, Figure 6.

Comment 4: Standard deviations (or errors) and significances should be added in Figure 2.

Response: We have modified accordingly in our revised manuscript and redrew the Fig 2. See page 7, Fig 2.

Comment 5: Why only one pollen unit was showed in Figure 4B?

Response: During our experiment, pollen viability determination showed that dwarf mutant has no pollen grain, three replications were just found one shriveled pollen, therefore, the male sterility of dwarfing mutant may also be caused by the lack of pollen.

Comment 6: Parameters of cells in Figure 5 should be quantitively described.

Response: Thanks for your valuable suggestions. The image J software was used to analyze data from Figure 5, the according modification in our revised manuscript. See page 8, line 283-288. Specific information was list in Table S3.

Comment 7: The quality of Figure 6 should be improved.

Response: We adjusted the quality of the Fig.6. See supplementary materials Figure S1.

Round 2

Reviewer 1 Report

I accept the corrections, so now in my opinion the manuscript is ready to be accepted for publication.

Reviewer 2 Report

The manuscript has been improved. I don't have any other questions.